# A Whole-Genome Sequencing-Based Approach for the Characterization of *Klebsiella pneumoniae* Co-Producing KPC and OXA-48-like Carbapenemases Circulating in Sardinia, Italy

**DOI:** 10.3390/microorganisms11092354

**Published:** 2023-09-20

**Authors:** Arcadia Del Rio, Valeria Fox, Narcisa Muresu, Illari Sechi, Andrea Cossu, Alessandra Palmieri, Rossana Scutari, Claudia Alteri, Giovanni Sotgiu, Paolo Castiglia, Andrea Piana

**Affiliations:** 1Department of Biomedical Science, University of Sassari, 07100 Sassari, Italy; delrio.arcadia2@gmail.com; 2Department of Oncology and Hemato-Oncology, University of Milan, 20122 Milan, Italy; valeria.fox@unimi.it (V.F.); rossana.scutari@unimi.it (R.S.); claudia.alteri@unimi.it (C.A.); 3Department of Humanities and Social Sciences, University of Sassari, 07100 Sassari, Italy; 4Department of Medicine, Surgery, and Pharmacy, University of Sassari, 07100 Sassari, Italy; illasechi@uniss.it (I.S.); andreacossu@uniss.it (A.C.); luca@uniss.it (A.P.); castigli@uniss.it (P.C.); piana@uniss.it (A.P.); 5Clinical Epidemiology and Medical Statistics Unit, Department of Medical, Surgical and Experimental Medicine, University of Sassari, 07100 Sassari, Italy; gsotgiu@uniss.it

**Keywords:** *Klebsiella pneumoniae*, whole-genome sequencing, carbapenemases, β-lactam/β-lactamases inhibitors

## Abstract

Background: Whole-genome sequencing (WGS) provides important information for the characterization, surveillance, and monitoring of antimicrobial resistance (AMR) determinants, particularly in cases of multi- and extensively drug-resistant microorganisms. We reported the results of a WGS analysis carried out on carbapenemases-producing *Klebsiella pneumoniae*, which causes hospital-acquired infections (HAIs) and is characterized by a marked resistance profile. Methods: Clinical, phenotypic, and genotypic data were collected for the AMR surveillance screening program of the University Hospital of Sassari (Italy) during 2020–2021. Genomic DNA was sequenced using the Illumina Nova Seq 6000 platform. Final assemblies were manually curated and carefully verified for the detection of antimicrobial resistance genes, porin mutations, and virulence factors. A phylogenetic analysis was performed using the maximum likelihood method. Results: All 17 strains analyzed belonged to ST512, and most of them carried the *bla*_KPC-31_ variant *bla*_OXA-48-like_, an OmpK35 truncation, and an OmpK36 mutation. Phenotypic analysis showed a marked resistance profile to all antibiotic classes, including β-lactams, carbapenems, aminoglycosides, fluoroquinolone, sulphonamides, and novel β-lactam/β-lactamase inhibitors (BL/BLI). Conclusion: WGS characterization revealed the presence of several antibiotic resistance determinants and porin mutations in highly resistant *K. pneumoniae* strains responsible for HAIs. The detection of *bla*_KPC-31_ in our hospital wards highlights the importance of genomic surveillance in hospital settings to monitor the emergence of new clones and the need to improve control and preventive strategies to efficiently contrast AMR.

## 1. Introduction

Antimicrobial resistance (AMR) represents one of the leading public health issues of the 21st century, with an estimated mortality rate of 10 million people per year by 2050 [1].

It has been estimated that, in 2019, AMR was directly responsible for nearly 1.3 million deaths, ranking only behind ischemic heart diseases and stroke in terms of fatalities. Almost one-quarter of these deaths were attributed to six of the most common pathogens: *Escherichia coli*, *Staphylococcus aureus*, *Klebsiella pneumoniae*, *Streptococcus pneumoniae*, *Acinetobacter baumannii,* and *Pseudomonas aeruginosa* [2].

The predominant mechanism of resistance involves the production of enzymes, such as carbapenemases, capable of binding and rendering antimicrobial drugs ineffective. Of particular concern is the rapid dissemination, both intra- and inter-species, of these enzymes through mobile genetic elements within nosocomial settings, resulting in a rapid spread among patients and hospital wards. This phenomenon is especially alarming given its significant impact on mortality and morbidity rates associated with hospital-acquired infections (HAIs). In addition to the challenges of managing and treating HAIs, there is also a substantial economic burden deriving from prolonged hospitalizations, the administration of new therapies, and the treatment of related complications [3].

Over the past few decades, *Klebsiella pneumoniae* has become the leading cause of opportunistic infections in healthcare facilities, causing a wide range of infections, including bloodstream, urinary, and respiratory tract infections [4]. In addition, the outcome of infections is complicated by the presence of both natural and acquired antimicrobial resistance determinants, which further limit therapeutic options and increase the risk of treatment failure. In Europe, carbapenem-resistant *Klebsiella pneumoniae* (cr-Kp) is common in eastern and southern nations, including Italy, where cr-Kp is endemic, and 29.5% of resistant isolates were detected in 2020 [5]. Several factors play a role in the dissemination of carbapenem-resistant bacteria in acute care hospitals or long-term facilities, including the number of colonized patients, the presence of comorbidities, the prolonged use of devices like central venous catheters, prior antibiotic use, and patient immunization status [6].

Effective monitoring and surveillance systems are, therefore, crucial to understanding global trends and planning coordinated actions to contrast antimicrobial resistance spread. To date, over 400 antimicrobial resistance genes have been identified in *K. pneumoniae*, with the number of related variants continuing to grow [7]. At the time of writing, a total of 17,456 *K. pneumoniae* whole genomes are present in the NCBI database, with a sequence length ranging from 5,000,000 to 7,000,000 bp [8]. Of these, 772 and 606 sequences carry *KPC* and *OXA* genes, respectively, with a sequence length ranging from 5,000,000 to 7,000,000 bp [9,10]. Limited information is available regarding strains carrying *KPC* and *OXA* genes simultaneously.

Traditional protocols for the diagnosis and surveillance of infectious diseases mainly rely on culture methods, which allow for microorganism identification and provide information on antibiotic susceptibility profiles. While these methods have been widely used because of their simplicity and the ability to precisely determine the minimum inhibitory concentration (MIC), which is particularly useful for the scheduling of treatments, they have some limitations. These limitations are primarily related to the length of the analysis time, typically from 24 to 72 h, which increases the risk of assigning empirical therapies and delays in the administration of proper antibiotics. Recently, molecular-based detection methods were considered the gold standard for the identification of β-lactamases, including carbapenemase-encoding genes, because of their high sensitivity and specificity and the significant reduction in analysis time [11]. Additionally, the use of whole-genome sequencing (WGS) for the monitoring and surveillance of antimicrobial resistance provides critical insights into the genomic characterization of multi- and extensively drug-resistant isolates (MDR and XDR, respectively), allowing for a better understanding of the genetic basis of antimicrobial resistance and virulence factors, as well as enabling the clarification of genomic epidemiology and the spatial–temporal evolution of strains in nosocomial settings [12].

As previously reported, the local epidemiological scenario is dominated by the widespread circulation of *K. pneumoniae* strains that co-produce class A and D carbapenemases, characterized by a pronounced resistance profile against all classes of antibiotics, including the new β-lactam–β-lactamase inhibitors (BL/BLIs) [13]. Moreover, the production of multiple carbapenemases has led to enhanced virulence and pathogenicity, increasing the risk of progressing from colonization to infection and raising the mortality rate among vulnerable patients [13]. Here, we aimed to analyze the genetic profile of KPC and OXA-48-like co-producing *K. pneumoniae* strains, collected in the University Hospital of Sassari (Sardinia, Italy) in a timespan of 1 year, and integrate this information with phenotypic and clinical data. The results obtained could prove useful in improving surveillance programs and planning effective preventive measures to counteract the spread of antimicrobial resistance isolates.

## 2. Materials and Methods

An observational monocenter study was carried out from December 2020 to December 2021 at the University Hospital of Sassari (Sardinia, Italy), within the local surveillance program for carbapenemase-producing *Enterobacteriaceae* (CPE). A set of multidrug-resistant *K. pneumoniae* KPC and OXA-48-like co-producing isolates, obtained from routine microbiological cultures of clinical specimens, were included in the present study. In detail, the present study includes all strains responsible for invasive bacterial infections (i.e., blood and broncholavage), classified as HAIs and characterized using a marked resistance profile to all antibiotic classes, including novel combinations of BL/BLIs.

A hospital-acquired infection is defined as an infection that occurs on or after the 3rd calendar day of admission to an inpatient location according to the National Healthcare Safety Network [14]. A case of infection sustained by *K. pneumoniae* is defined by the isolation of the strain from a clinically significant sample, with the simultaneous occurrence of clinical signs of infection caused by the isolated microorganism. Biological specimens considered clinically significant were blood and bronchial lavage cultures. Conversely, colonization status was defined as the isolation of *K. pneumoniae* from the rectal swab of a patient without clinical signs or symptoms of infection [15].

Only the strains isolated for the first time from a given patient were included. Patients’ data (i.e., age, sex), type of specimens, clinical outcome, date of detection, and ward of admission were recorded and reported in an ad hoc file.

### 2.1. Microbiological Identification and Phenotypic/Genotypic Characterization of K. pneumoniae Strains

Clinical specimens were plated onto nutritive and selective media agar for microbiological growth, including selective chromogenic media for screening carbapenemase-producing *Enterobacteriaceae* (Chromid Carba Smart Agar—bioMérieux, Grassina, Italy). After overnight incubation in aerobic conditions at 35–37 °C, the suspected bacterial colonies were further analyzed for identification and antimicrobial susceptibility testing.

A fully automated analysis system (Vitek II) was used for species identification and antimicrobial susceptibility testing, using the GN ID card and the AST-N379 card, respectively, including the following antibiotics: amikacin, amoxicillin/clavulanic acid, piperacillin/tazobactam, cefepime, cefotaxime, ceftazidime, ceftolozane/tazobactam, meropenem, imipenem, gentamicin, tobramycin, ciprofloxacin, trimethoprim/sulfamethoxazole [16].

Simultaneously, E-test assays were carried out to verify the susceptibility for the new BL/BLI, which were not included in the available Vitek AST cards, using E-test gradient strips (bioMérieux, Inc., Durham, NC, USA) for ceftazidime/avibactam (CZA), imipenem/relebactam (I/R) and meropenem/vaborbactam (M/V) as follows: A 0.5 McFarland suspension in 0.85% sterile saline was inoculated onto Mueller–Hinton agar plates with a sterile cotton swab; subsequently, E-test strips were applied to plates manually. After incubation for 16–20 h at 37 °C, the plates were read. The results were interpreted according to the clinical breakpoints of the European Union Committee on Antimicrobial Susceptibility Testing, EUCAST v.13.0 [17], in order to assess MIC (minimal inhibitory concentration) values.

A total of 17 *K. pneumoniae* strains that exhibited a MIC value equal to or higher than EUCAST values were defined as carbapenem-resistant and underwent molecular analysis for the detection of the most common antimicrobial resistance determinants. In detail, the following carbapenemase-encoding genes were detected via real-time PCR, *bla*_KPC_*, bla*_NDM_*, bla*_VIM_*, bla*_IMP_, and *bla*_OXA-48_, using the commercial Allplex Entero DR assay kit [18]. Simultaneously, isolates were tested for the production of extended-spectrum β-lactamases (ESBLs) through the detection of *bla*_CTX-M_.

### 2.2. Whole-Genome Sequencing and Bioinformatics Analysis

Bacterial strains characterized by a marked resistance profile to all antibiotic classes, including BL/BLIs, responsible for invasive infections and isolated during the time of observation, were subsequently analyzed with WGS.

After an overnight culture of the strains at 37 °C, genomic DNA was extracted using a genomic DNA extraction kit (QIAamp DNA Kit, QIAGEN, Hilden, Germany) and sequenced using the Illumina NovaSeq 6000 platform (Illumina, San Diego, CA, USA) with a pair-end strategy of 150 bp to obtain an average 50× sequencing depth.

Raw reads were trimmed for adapters and filtered for quality (average quality > 20) with Fastp (v0.20.1) [19]. FastQC (v0.11.9) was used for quality checks after trimming [20]. A de novo genome assembly was performed using the SPAdes Genome Assembler (v3.14.1) using the “-careful” option [21]; the quality of the assemblies was evaluated with Quast (v5.1). Annotation of the assembled contigs was performed with Prokka (v1.14.6) [22], and isolate typing was performed with the MLST tool (v2.11) [23].

Kaptive (v0.7.3) and Kleborate (v2.2.0) were used to predict the polysaccharide capsular (K) and lipopolysaccharide O antigen (O) loci, antimicrobial resistance determinants, and virulence factors [24].

MobileElementFinder tool (v1.0.3) and MOB-suite (v3.1.2) were used for mobile genetic elements and plasmid identification, respectively [25,26,27]. The presence of Tn4401 transposons, their type, and their flanking sequences were detected using TETyper (v1.1) [28].

The *bla*_KPC_ gene copy number was estimated with the ccne tool (v1.1.0) [29]. The presence of a mixed population of *bla*_KPC_ variants was investigated by aligning raw reads to the *bla*_KPC-31_ reference sequence (CARD accession ARO:3005362) with the bwa-mem algorithm (v0.7.17) and by identifying the single-nucleotide polymorphisms (SNPs) with SAMtools (v1.4) and freebayes (v1.3.2) [30].

### 2.3. Phylogenetic Analysis and Bacterial Typing

Two phylogenetic trees were constructed from the core gene alignments obtained with Roary (v3.13.0) [31] with default parameters, obtaining a core gene alignment shared by 95% of the isolates.

The first tree, comprising 11 reference genomes (5 for ST512, 2 for ST147, 2 for ST307, 1 for ST20, and 1 for ST101) plus the sequences obtained in this study, was used to confirm the ST assignment. A second phylogenetic tree, comprising only the strains obtained in this study, was performed in order to highlight potential transmission clusters.

The maximum likelihood (ML) phylogeny was estimated with IQTREE (v2.0.6) [32] using the best-fit models of nucleotide substitution, TIM + F + I and HKY + F + I, both inferred using ModelFinder [33] with 1000 bootstrap replicates.

Phylogenetic trees were visualized and annotated using iTOL (v5) [34].

To confirm sequence segregation, a minimum spanning tree was constructed on the core genome alignment obtained from the study strains with Grapetree (v1.5.0). Pairwise SNP distances were calculated using the snp-dists tool [35], and SNP localization and variants were evaluated with the Gingr Harvest Suite [36]. Pairwise genetic distance was calculated for the core genome with MEGA (v6).

### 2.4. Data Availability

The 17 *Klebsiella pneumoniae* sequence data obtained in this study are openly available on the European Nucleotide Archive (ENA) under accession no. ERS15409321–ERS15409337.

## 3. Results

Based on the previously mentioned criteria, a total of seventeen strains, responsible for HAIs, were included in this study. Clinical specimens were mainly obtained from blood cultures (58.8%; 10/17), followed by broncholavage (35.3%; 6/17), whereas only one strain was isolated from the rectal swab of a patient who concomitantly developed colonization and infection status and for which the genotypic characterization showed a unique profile.

The mean (SD) patient age was 65 years (±10.8); the majority of them were male (70.6%; 12/17), without any statistical differences in age between genders. Data from medical records, collected with CPE screening, showed that positive rectal swabs were previously detected in all patients and that the median (IQR) number of days from colonization to infection was 16 (4.25–20.5) days. Most of the hospital admissions are attributable to complications due to SARS-CoV-2 infection (58.8%; 10/17). During the period of this study, the mortality rate was 65.7% (11/17), without any statistically significant difference between COVID-19-positive or -negative patients.

Three-quarters of the patients were hospitalized in the intensive care unit (ICU) (76.5%; 13/17) and 11.8% (2/17) in the pneumology ward, followed by the medicine and the long-term care units, which accounted for one strain each (5.8%; 1/17).

### 3.1. Microbiological Identification and Characterization

Phenotypic analysis revealed that all strains belonged to the *Klebsiella pneumoniae* species, whereas the genotypic characterization, carried out using real-time PCR, showed the co-expression of *bla*_KPC_ and *bla*_OXA-48-like_ carbapenemase-encoding genes in all strains.

Antimicrobial susceptibility testing showed a pronounced resistance profile to all antibiotic classes, including β-lactams, aminoglycosides, fluoroquinolones, and sulphonamides (Table 1). Moreover, all strains were resistant to carbapenems, with median MIC values of 16 µg/mL and 8 µg/mL for meropenem and imipenem, respectively. We found a marked resistance profile for new BL/BLI combinations (i.e., CZA, I/R, and M/V) for all strains: 94.2% (16/17) of strains reported a MIC value > 256 µg/mL for CZA, and the median MIC value for I/R was 4 µg/mL, whereas for M/R, the median MIC value observed was 8 µg/mL. Based on the EUCAST Clinical Breakpoint [17], all strains were classified as resistant to CZA and M/V, whereas only one strain was sensitive to the I/R combination.

### 3.2. Whole-Genome Sequencing and Bioinformatics Analysis

The median (IQR) of reads obtained after filtering was 11.07 (10.16–12.00) Mb. The whole-genome assemblies of the isolates displayed a median (IQR) number of contigs of 250 (237–259), with a median (IQR) N50 of 157,591 (154,488–165,249) bp, while the total isolate genome size ranged from 5.76 to 5.89 Mb, with a mean GC content of 57.01% and a median (IQR) number of predicted genes of 5571 (5567–5577) (Appendix A).

An analysis of resistance genes with Kleborate showed the concomitant presence of genes conferring resistance to several antibiotic classes, like aminoglycosides, fluoroquinolone, macrolides, chloramphenicol, rifampicin, sulphonamides, tetracyclines, trimethoprim/sulfamethoxazole, and β-lactams/carbapenems. Among the known antimicrobial resistance genes involved in β-lactam resistance, we found the consistent presence of *bla*_OXA-181_*, bla*_OXA-9_*, bla*_OXA-10_*, bla*_CMY-59_, and *bla*_SHV-11_ genes in all strains, with only the exception of *bla*_TEM-122_, which was detected in 88.2% (15/17) of isolates. Moreover, the sequencing analysis evidenced the presence of several genes already described as the most common mechanisms involved in antibiotic resistance (Table 2 and Appendix A).

All strains carried *bla*_KPC_ genes; moreover, two different variants were identified: *bla*_KPC-31_, detected in the majority of strains (88.2%, 15/17), and *bla*_KPC-3_, detected in two strains (A6 and A8).

In line with phenotypic resistance tests, strains carrying *bla*_KPC-31_ displayed full resistance to CZA (MIC: 256 µg/mL). Of note, all strains were also resistant to meropenem, notwithstanding the ability of the KPC-31 enzyme to restore carbapenem susceptibility [37,38]. In order to investigate the potential mechanisms underlying the resistance to meropenem found among *bla*_KPC-31_ isolates, we assessed the presence of truncations and/or mutations in outer membrane protein porins (Omps), already known to be associated with a loss of carbapenem activity [39,40], and we found that all strains carrying *bla*_KPC-31_ carried a truncated OmpK35 and a mutated OmpK36 (OmpK36GD mutation).

Regarding the two strains carrying *bla*_KPC-3_ (A6 and A8), phenotypic resistance tests defined a CZA MIC of eight and 256 µg/mL, respectively. In order to investigate potential mechanisms underlying the increased MIC value for CZA, we first estimated the *bla*_KPC-3_ copy number with the ccne tool and, subsequently, evaluated the potential presence of a mixed *bla*_KPC_ population by aligning the raw reads of the two samples against the *bla*_KPC-31_ gene. The analysis using the ccne tool did not reveal an increase in the copy number of the *bla*_KPC_ gene, whereas strain A8 showed a mixed population of *bla*_KPC-3_ (55.7% of mapping reads) and *bla*_KPC-31_ (44.3%).

### 3.3. Phylogenetic Analysis and Bacterial Typing

The MLST analysis revealed that all strains belonged to the same sequence type (ST), 512, and carried the same capsular (KL) and lipopolysaccharide (O) loci, respectively, KL107 and O2afg. The maximum likelihood (ML) phylogenetic tree, inferred from a core genome alignment of 3,948,089 bp, confirmed the distribution of strains based on the STs (Appendix A).

When considering only the 17 strains obtained in this study, we found that they were characterized by a median (IQR) genetic distance of 8.4 × 10^−6^ (4.4 × 10^−6^–1.2 × 10^−5^) base substitutions per site (Figure 1).

Only one cluster with a bootstrap > 90% was observed, which comprised strains A2, A3, A10, and A17. Data reported in clinical records showed that all cases were isolated between March and April 2021 and that three out of these four cases (A3, A10, and A17) were diagnosed among patients hospitalized in the intensive care unit. This cluster had a median (IQR) number of single-nucleotide polymorphisms (SNPs) of 11 (8–13) (Figure 2). Of these SNPs, some were found to be concentrated in particular genomic regions, including three SNPs falling into insertion sequences (A2370833T, C2371047T, and G2371072A, with respect to the core genome alignment), two falling into genes involved in the cell wall biosynthesis (G3168718A, peptidoglycan DD-transpeptidase MrdA, and G3169291A in the penicillin-binding protein PBP-2), six in the aminoglycoside O-phosphotransferases APH(3′)-Ia and APH(3″)-Ib (C3296233A, A3296320G, A3296408C, G3296667T, and G3296993A in APH(3′)-Ia and A3295925T in APH(3″)-Ib), and two in elongation factor Tu (C4461555T and T4461570C). The other SNPs identified in this cluster were mostly interspersed throughout the core genome and included, among others, one SNP falling into the pilus assembly protein (A2366425G, with respect to the core genome alignment) and one in the pyrroloquinoline quinone biosynthesis protein B (T3724855C), together with other SNPs falling into genes involved in metabolic processes.

Regarding the other strains, they did not cluster together in the ML tree and were found to be interspersed along the MS tree, with a median (IQR) number of SNPs of 28 (14–40) (Figure 2), albeit displaying the same ST, K, and O loci and overall carrying the same antibiotic resistance genes (Figure 1).

All *bla*_KPC_ genes identified were found to be carried by a Tn*4401*a transposon, always present as a single copy. Among the other transposons identified, Tn*6196* was found in all strains, while Tn*801* was found in 35.3% (6/17) of strains (Appendix A).

A plasmid analysis assessed the presence of 11 different plasmids, homologous to plasmids deposited in GenBank, some of which carried most of the resistance genes (Appendix A).

Besides the core factors naturally present in *K. pneumoniae* and enterobactin, no other virulence factors were identified (Appendix A).

## 4. Discussion

In the present study, we report the phenotypic and genotypic characteristics of *K. pneumoniae* strains responsible for HAIs, isolated by the antimicrobial resistance surveillance program of the University Hospital of Sassari (Italy) between December 2020–December 2021.

According to international guidelines, a protocol for monitoring antibiotic resistance has been active in our hospital since 2015. As already described in previous reports [13,41], the local epidemiological scenario has shown an increased variability in terms of circulating variants, particularly among carbapenemase-producing Gram-negative bacteria, such as *Escherichia coli bla*_VIM_ and *Enterobacter* spp. class A carbapenemase producers. Recently, molecular surveillance performed on rectal swabs of hospitalized patients identified *K. pneumoniae* strains, co-producing class A and D carbapenemases, as the main cause of colonization in our setting, resulting in an alarming rise in invasive infections. Genomic characterizations of these isolates, coupled with analyses of clinical and microbiological data, become crucial for effectively planning preventive measures and monitoring the trend of colonization/infection cases within hospital wards.

Findings from medical records have evidenced several factors potentially involved in the progression from colonization to infection, including the mean age of patients, the prior ascertainment of colonization status, the admitting ward, and the length of stay [42]. These findings underscore the importance of identifying risk factors involved in the development of HAIs and the necessity of implementing targeted interventions on patients at high risk. Additionally, more than half of our patients tested positive for SARS-CoV-2 upon hospital admission. The impact of the COVID-19 pandemic on AMR has been described by several studies, also taking into consideration the frequent co-occurrence of bacterial and SARS-CoV-2 infections. Among COVID-19 patients, mechanical ventilation, severe immunosuppression, and the administration of broad-spectrum antibiotics are frequently observed and might significantly contribute to the development of HAIs. Moreover, the concomitant administration of antibiotic therapy on these patients has further increased the selective pressure in nosocomial settings, facilitating the emergence of new MDR strains [43].

The phenotypic analysis revealed a pronounced resistance profile in all strains included in the study, encompassing the most common antibiotic classes, including novel β-lactam/β-lactamase inhibitor combinations (i.e., CZA, M/V, I/R). The high mortality rate (>65%) registered in our cohort highlights the risk of limited therapeutic success in cases of severe infection caused by MDR strains. The presence of cross-resistance, defined as the ineffectiveness of all antibiotics within the same class due to a single mechanism [44], is of particular concern, as it restricts treatment options and consequently hampers favorable outcomes. The lack or delay in administering proper therapy could contribute to a negative outcome. Indeed, studies have shown that the time before appropriate antibiotic therapy is a strong predictor of 30-day mortality in patients with bloodstream infections caused by KPC-producing *K. pneumoniae* [3].

Since its first detection in 2001, the prevalence of KPC-producing *K. pneumoniae* isolates has rapidly increased worldwide because of the presence of *bla*_KPC_ in mobile genetic elements, which facilitate its dissemination among bacterial species [45]. Previous surveillance studies have reported a clear prevalence of *bla*_KPC-2_ variants globally, whereas a predominance of *bla*_KPC-3_ has been observed in Italy [46]. AMR-monitoring programs in our region have shown that KPC expression is the primary mechanism of resistance in *K. pneumoniae*, as well as in other *Enterobacteriaceae*. In this study, sequencing analyses revealed that all strains belonged to ST512 and carried two different variants of *bla*_KPC_: *bla*_KPC-3_ in two strains (A6 and A8) and *bla*_KPC-31_ in the remaining fifteen strains. Although the *bla*_KPC-3_ gene is not commonly associated with BL/BLI resistance, both A6 and A8 showed medium and high levels of CZA resistance, respectively. Among the mechanisms known to be involved in CZA resistance, the presence of porin mutations, an increase in the *bla*_KPC_ copy number, and the occurrence of a mixed population of *bla*_KPC_ genes have also been reported [47,48,49,50,51]. The detection of a mixed population of *bla*_KPC-3_ and *bla*_KPC-31_ in the A8 strain can explain the high MIC value detected for CZA (MIC = 256 µg/mL). As previously noted, the presence of different subpopulations could negatively predict treatment outcomes and increase the risk of selecting multidrug-resistant clones [50].

The detection of the *bla*_KPC-31_ variant, found in 15 isolates, raises additional concerns given its ability to confer CZA resistance as a consequence of a mutation within the Ω-loop (amino acid positions 163–179), which surrounds the active site of the protein determining an increased binding affinity toward ceftazidime and a reduction in sensitivity to avibactam [52]. To date, more than 20 *bla*_KPC_ genes have been deposited in GenBank, with many characterized by mutations in the Ω-loop region, leading to CZA resistance. KPC-31, which represents a KPC-3 variant, is widely reported globally and was recently described in an ST512 isolate in Italy [53]. Interestingly, this variant is often described as being able to confer a CZA-resistant phenotype, while simultaneously increasing susceptibility to carbapenems [39]. However, all the strains included in our study exhibited both BL/BLIs and carbapenem resistance, likely because of the presence of several AMR determinants. For instance, it is well known that a decrease in or loss of function in outer membrane proteins can affect carbapenem activity in *K. pneumoniae*. The detection of truncations and mutations in outer membrane protein porins among our isolates could explain the persistence of carbapenem resistance. Furthermore, avibactam, relebactam, and vaborbactam rely on OmpK35 and OmpK36 to cross the bacterial outer membrane, and thus, changes in membrane permeability could result in an increase in median MICs by 2-, 4-, and 16-fold for CZA, I/R, and M/V, respectively [54].

In summary, there is a need to gain a better understanding of the role of KPC variants in determining AMR phenotypes, especially when considering new combinations of antimicrobial drugs. Additionally, several studies have highlighted the risk of induced resistance to CZA following treatment [55]. Therefore, the emergence of resistant clones due to selective pressure highlights the importance of implementing effective antimicrobial stewardship strategies.

All strains were found to carry *bla*_OXA-181_, which differs from *bla*_OXA-48_ by four amino acid substitutions and shares the same β-lactam hydrolysis activity against penicillin and carbapenems but does not confer resistance to extended-spectrum cephalosporins [56]. The number of *K. pneumoniae*-encoding OXA-48-like isolates has been increasing, and *bla*_OXA-181_ is now becoming one of the most common class D carbapenemases worldwide, mainly because of its plasmid location, which enables its spread via horizontal transfer [57]. This feature can explain the rapid diffusion of *K. pneumoniae* strains co-producing KPC and OXA-48-like carbapenemases within our nosocomial setting. Of note, our report is one of the first genomic investigations at a national level of *K. pneumoniae* strains co-harboring the *bla*_KPC_ and *bla*_OXA-48-like_ genes. Similar isolates were detected and reported in Taiwan and India by [58,59]. Both studies highlighted the rapid spread of these bacteria within hospital wards and their pronounced resistance profile, which significantly limits the available therapeutic choices.

The localization of AMR determinants in mobile genetic elements facilitates the emergence of MDR clones and, concurrently, allows for the acquisition of new resistance genes, further exacerbating the resistance profile [60]. Moreover, we found the Tn4401 transposon—which is able to drive most of the resistance genes and is known to be one of the main factors responsible for the rapid dissemination of KPC-producing *K. pneumoniae* worldwide—to be carried by plasmids. Eight isoforms (a to h) of Tn4401 have been identified so far, differing for deletions immediately upstream of the *bla*_KPC_ gene, with Tn*4401*a and Tn*4401*b being the most common [61]. In our study, Tn*4401*a was identified in all strains carrying both the *bla*_KPC-3_ and *bla*_KPC-31_ genes, whereas ColKP3 and IncX3 are reported to be the main promoters involved in the spread of the *bla_OXA-181_* and *qnrS1* genes, conferring resistance to carbapenems and fluoroquinolones, respectively [62]. The presence of different AMR determinants within the same mobile elements highlights the genetic diversity of plasmids and their key role in the diffusion of AMR.

All strains involved in the present study belonged to ST512, already reported to be the main ST in our epidemiological scenario [63]. Our data are consistent with national and international reports, which describe ST512 as one of the most prevalent sequence types worldwide, clustered into clonal group (CG) 258, together with ST11 and ST258, which are considered to be its ancestors [64]. Strains belonging to CG 258 are recognized as “high-risk” clones because of the following characteristics: (i) their presence and isolation from various international geographical locations; (ii) a multidrug resistance profile; (iii) enhanced fitness, virulence, and pathogenicity; (iv) an ability to establish host colonization; and (v) extensive transmission between hosts. For these reasons, strict surveillance within hospital settings is imperative [65].

The ability of ST512 *K. pneumoniae* strains to survive and spread within hospital wards might be a key factor contributing to their significant prevalence in our setting. Strict genomic surveillance and the implementation of infection control practices, such as the isolation of colonized patients and hand hygiene, are needed to effectively track and contain cr-Kp transmission between hospitalized patients.

While the phenotypic profile revealed an extensive resistance to all antibiotic classes, the genotypic analysis indicated only the presence of enterobactin, the most common virulence factor known in *K. pneumoniae*. We also observed the presence of capsular (K) and lipopolysaccharide (O) loci, both known to influence pathogenicity in *K. pneumoniae* and to play roles in immune evasion mechanisms. In particular, the K-antigens, present in the capsular polysaccharide layer, can completely evade the host immune response, reducing the adhesion of phagocytes to epithelial cells. Although the O-antigen in lipopolysaccharide can vary based on the structural composition and glycosidic linkages, the poor ability of O2afg-antigens to induce inflammatory responses facilitates bacteria survival and dissemination within the host [66,67].

The only cluster identified via sequencing analysis mainly involved patients hospitalized in the ICU for a narrow period, highlighting the strong correlation between infection events and clinical practices. Conversely, the variability observed in the mobile genetic elements highlights the bacterial ability to easily acquire new determinants and rapidly spread among vulnerable patients [58]. Based on previous reports, which showed an increased AMR rate during the COVID-19 pandemic, particularly for Gram-negative bacteria, we hypothesize that the concomitant spread of SARS-CoV-2 may have partially facilitated the rapid dissemination of these MDR strains. This highlights the crucial need for strengthened infection prevention and control measures and enhanced antimicrobial stewardship in the post-pandemic era [68,69].

## 5. Conclusions

Overall, despite the relatively small number of strains examined, the present study revealed the circulation of *K. pneumoniae* strains co-expressing *bla*_OXA-48-like_ and *bla*_KPC-31_ in our wards, the latter being able to confer resistance to various antibiotics, including new BL/BLIs, further limiting our available therapeutic options.

The present work, although providing only a partial description of the local epidemiological context, underscores the importance of integrating phenotypic and genotypic analyses into AMR surveillance programs. A detailed genomic characterization of circulating *K. pneumoniae* strains is indeed essential for accurately identifying carbapenemase variants and potentially predicting their antibiotic susceptibility profiles.

These findings also reinforce the critical need to rigorously implement standard precautions in all care settings, such as hand hygiene, the use of gloves, eye and respiratory protection, and isolation precautions for colonized/infected patients, to significantly reduce the spread of MDR bacteria.

## Figures and Tables

**Figure 1 microorganisms-11-02354-f001:**
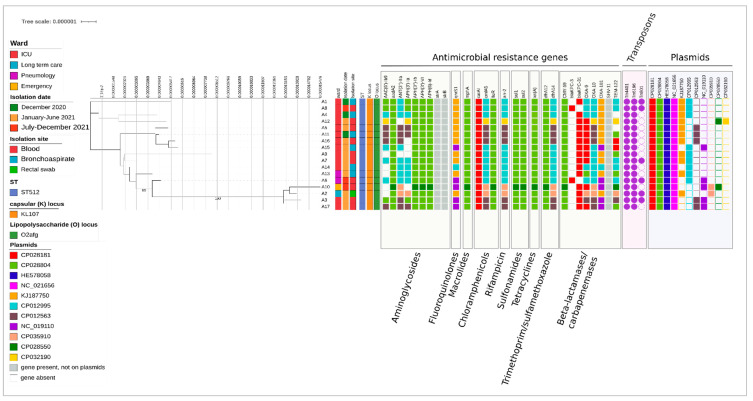
Estimated maximum likelihood phylogenetic analysis of *Klebsiella pneumoniae* isolates (n = 17). The phylogeny was estimated based on a core genome alignment of 5,004,681 bp with IQTREE using the best-fit model of nucleotide substitution, HKY + F + I, with 1000-replicate fast bootstrapping. The lefthand numbers represent the sample IDs; bootstrap values higher than 80 are shown on branches. Antimicrobial resistance genes are colored based on the plasmid on which they were identified. When they are not found on plasmids, they are colored gray, whereas the absence of genes is reported with blank-filled squares. The tree scale is expressed as nucleotide substitutions per site.

**Figure 2 microorganisms-11-02354-f002:**
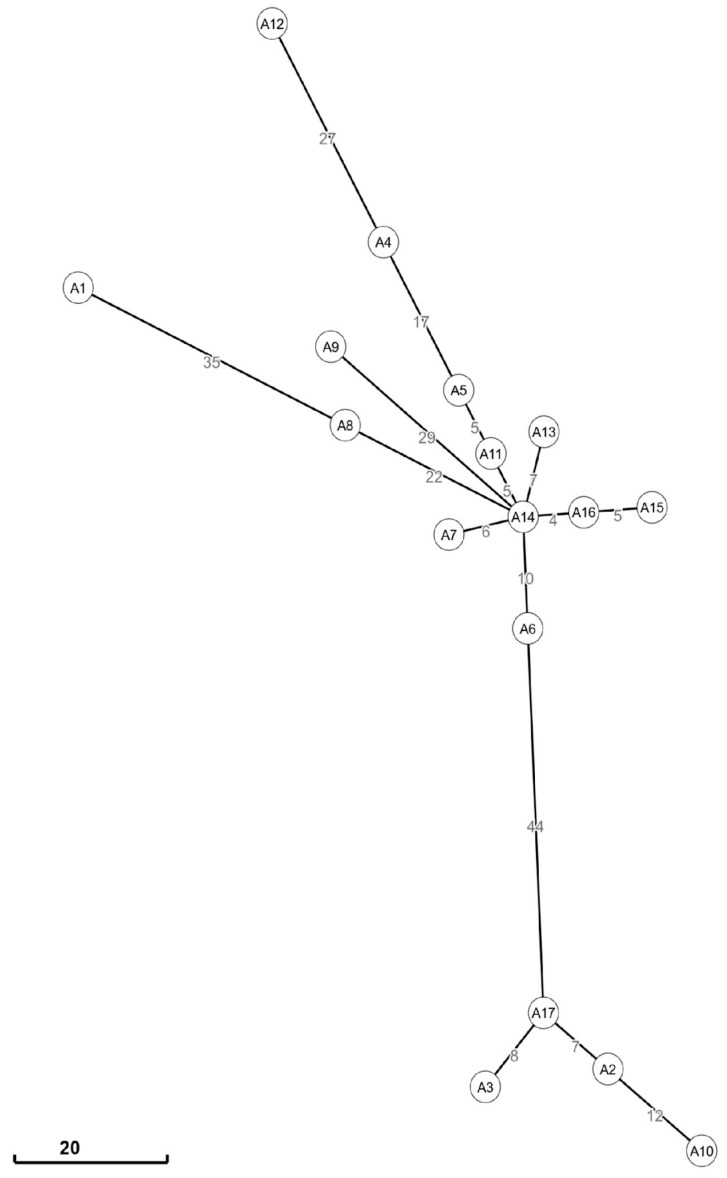
Minimum spanning trees displaying the pairwise SNP distances of *Klebsiella pneumoniae* isolates (n = 17). The minimum spanning tree was estimated with Grapetree based on a core genome alignment of 5,004,681 bp. ID nodes represent the sample IDs, while pairwise SNP distances between samples are reported as numbers on branches.

**Table 1 microorganisms-11-02354-t001:** Number and percentages of resistance strains and their relative minimum inhibitory concentrations against antibiotics.

Antibiotic Class	Antibiotic	Resistant Strains, n (%)	Median (IQR) MIC (µg/mL)
β-lactams	*Amoxicillin/clavulanic acid*	17 (100)	32 (32–32)
	*Piperacillin/tazobactam*	17 (100)	128 (128–128)
	*Cefepime*	17 (100)	32 (32–32)
	*Cefotaxime*	17 (100)	64 (64–64)
	*Ceftazidime*	17 (100)	64 (64–64)
	*Ceftolozane/tazobactam*	17 (100)	32 (32–32)
Carbapenems	*Meropenem*	17 (100)	16 (16–16)
	*Imipenem*	16 (94.1)	8 (8–8)
Aminoglycosides	*Amikacin*	17 (100)	32 (32–64)
	*Gentamicin*	17 (100)	16 (16–16)
	*Tobramycin*	17 (100)	16 (16–16)
Fluoroquinolone	*Ciprofloxacin*	17 (100)	4 (4–4)
Sulphonamides	*Trimethoprim/Sulfamethoxazole*	17 (100)	320 (320–320)
β-lactam–β-lactamase inhibitors	*Ceftazidime/avibactam*	17 (100)	256 * (256–256)
	*Imipenem/relebactam*	17 (100)	4 * (3–4)
	*Meropenem/vaborbactam*	17 (100)	8 * (8–12)

* Assessed with E-test^®^. MIC = minimum inhibitory concentration. IQR = interquartile range.

**Table 2 microorganisms-11-02354-t002:** Number and percentages of strains carrying resistance genes, grouped according to the corresponding antibiotic classes.

Antibiotics	Resistance Genes	N (%) of Strains
Aminoglycosides	*aac(6*’*)-Ib9*	12 (70.6)
	*aadA2*	17 (100)
	*ANT(3*″*)-IIa*	15 (88.2)
	*APH(3*’*)-Ia*	13 (76.5)
	*APH(3*″*)-Ib*	17 (100)
	*APH(3*’*)-VI*	17 (100)
	*APH(6)-Id*	17 (100)
	*strA*	17 (100)
	*strB*	17 (100)
Fluoroquinolones	*qnrS1*	17 (100)
Macrolides	*mphA*	17 (100)
Chloramphenicol	*catA1*	17 (100)
	*cmlA5*	17 (100)
	*floR*	17 (100)
Rifampicin	*arr-2*	17 (100)
Sulphonamides	*sul1*	17 (100)
	*sul2*	17 (100)
Tetracyclines	*tet(A)*	17 (100)
Trimethoprim/sulfamethoxazole	*dfrA12*	17 (100)
	*dfrA14*	17 (100)
Beta-lactams	*bla* _CMY-59_	17 (100)
	*bla* _KPC-3_	2 (11.7)
	*bla* _KPC-31_	15 (88.2)
	*bla* _OXA-9_	17 (100)
	*bla* _OXA-10_	17 (100)
	*bla* _OXA-181_	17 (100)
	*bla* _SHV-11_	17 (100)
	*bla* _TEM-122_	15 (88.2)

## Data Availability

The data is available upon specific request.

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
