# Peer review of "A Whole-Genome Sequencing-Based Approach for the Characterization of Klebsiella pneumoniae Co-Producing KPC and OXA-48-like Carbapenemases Circulating in Sardinia, Italy"

_microorganisms, 2023, doi:10.3390/microorganisms11092354_

Round 1

Reviewer 1 Report

Please see the attached file for the Reviewers Report on the manuscript microorganisms-2573356. Thank you!

Extensive editing of English language required.

Reviewer 2 Report

This study numbers of Klebsiella pneumoniae was isolated and sequenced. Further, the phylogenetic analysis, antimicrobial resistance genes and phenotype were analyzed. It is meaningful work. Some revisions are suggested to be drawn before accepted for publication.

(1)Title of the manuscript and Lines 42-44,the strain name should be in italic, please check all the manuscript

(2)No Figure 3 was found in the manuscript, but mentioned at lines 253, 268

(3)The Figure quality should be improved.

(4)In the introduction part, the risk of KPC and OXA-48 co-producing should be further described.

(5) Did the KPC and OXA-48 co-producing Klebsiella pneumoniae also found in other places of Italy or other country.

(6) How about the expression ratio of KPC and OXA-48 compared with other studies.

Reviewer 3 Report

I have read with interest the manuscript submitted by Del Rio et al, since AMR represents a global concern.

I have just a few comments to be addressed, in order to improve the quality of the manuscript:

- Once you described an abbreviation, use the abbreviation form only (ex row41)

- rows 42-44,54,59 etc. - italicize the bacteria names

- Can you mention if there were differences in mortality rates for patients with/without COVID-19 infections?

- In the studied period, there were other types of carbapenemases identified? It is worth at least mentioning, even though you focused on the characterization of KPC and OXA-producing KP. Any other bacteria, apart from KP that produced carbapenemases?

- what was the resistance rate to Colistin? 

The main limitation of the study is the low number of strains included.

Maybe consider including some more references, including studies that focused on K. pneumoniae`s resistance pattern and carbapenemase production before and after the onset of the pandemic.

minor spell/punctuation errors.

Round 2

Reviewer 1 Report

Please see the attached file for the Second Round of the Reviewers Report on the microorganisms-2573356. Thank you.

Moderate editing of English language required for the microorganisms-2573356.
